# Non-Traditional Starches, Their Properties, and Applications

**DOI:** 10.3390/foods12203794

**Published:** 2023-10-16

**Authors:** Evžen Šárka, Andrej Sinica, Petra Smrčková, Marcela Sluková

**Affiliations:** Department of Carbohydrates and Cereals, University of Chemistry and Technology, Prague, Technicka 5, 166 28 Prague, Czech Republic; sinicaa@vscht.cz (A.S.); jankovsp@vscht.cz (P.S.); mcerna@vscht.cz (M.S.)

**Keywords:** starch analyses, granule size, resistant starch, nanomaterials, edible films, amylose–lipid complexes, temperature of gelatinization

## Abstract

This review paper focuses on the recent advancements in the large-scale and laboratory-scale isolation, modification, and characterization of novel starches from accessible botanical sources and food wastes. When creating a new starch product, one should consider the different physicochemical changes that may occur. These changes include the course of gelatinization, the formation of starch–lipids and starch–protein complexes, and the origin of resistant starch (RS). This paper informs about the properties of individual starches, including their chemical structure, the size and crystallinity of starch granules, their thermal and pasting properties, their swelling power, and their digestibility; in particular, small starch granules showed unique properties. They can be utilized as fat substitutes in frozen desserts or mayonnaises, in custard due to their smooth texture, in non-food applications in biodegradable plastics, or as adsorbents. The low onset temperature of gelatinization (detected by DSC in acorn starch) is associated with the costs of the industrial processes in terms of energy and time. Starch plays a crucial role in the food industry as a thickening agent. Starches obtained from ulluco, winter squash, bean, pumpkin, quinoa, and sweet potato demonstrate a high peak viscosity (PV), while waxy rice and ginger starches have a low PV. The other analytical methods in the paper include laser diffraction, X-ray diffraction, FTIR, Raman, and NMR spectroscopies. Native, “clean-label” starches from new sources could replace chemically modified starches due to their properties being similar to common commercially modified ones. Human populations, especially in developed countries, suffer from obesity and civilization diseases, a reduction in which would be possible with the help of low-digestible starches. Starch with a high RS content was discovered in gelatinized lily (>50%) and unripe plantains (>25%), while cooked lily starch retained low levels of rapidly digestible starch (20%). Starch from gorgon nut processed at high temperatures has a high proportion of slowly digestible starch. Therefore, one can include these types of starches in a nutritious diet. Interesting industrial materials based on non-traditional starches include biodegradable composites, edible films, and nanomaterials.

## 1. Introduction

Starch makes up almost 90% of the daily calorie intake from staple foods worldwide [1]. It affects food texture and is used in non-food applications.

This carbohydrate is industrially isolated mainly from corn, tapioca (cassava), wheat, potato, sago [2], pea, arrowroot [3,4], and rice. The two most common starch sources are corn (75% of global production) and tapioca (12%). In Europe, starch is manufactured mainly from corn, wheat, and potato. Additionally, pea starch is produced by Emsland Group (Germany) and by Roquette Frères (France). Oat starch is also widely used in the pharmaceutical and cosmetic industries [5]. Finland is one of the few countries in the world producing barley starch. During World War II and later in the former German Democratic Republic, ray starch was manufactured [1].

Several commercial starches originate from specific cultivars containing high-amylose starch (70% amylose) or waxy starch with a high amylopectin content. Amylose corn starch is produced by Xinfuwang New Material Technology Co., Ltd. (Henan, China) [6], and waxy corn starch by Lai Tangyuan Food Co., Ltd. (Jiangsu, China) [7] or by Samyang Genex Company (Seoul, Republic of Korea) [8].

Some starches are practically unknown in Europe. Achira starch (*Canna edulis*) is produced in Colombia and Venezuela [4,9], and Dua Naga, Ltd. (Indonesia) provides arenda starch. Lotus root starch is manufactured by Yangzhou Lianshun Food Co., Ltd. (Yangzhou, China) [10]. Mung bean starch can be purchased from Hengshui Fuqiao Starch Co., Ltd. (Jingxian, China) [11]. In the text below, there will be some discussion about sago and arrowroot starch.

Starch has many applications in food and non-food industries. However, the use of starch in its original form has some restrictions due to a poor flow ability, low paste transparency, susceptibility to retrogradation after gelatinization, high syneresis, low shear stress resistance, high gelatinization temperatures, rapid enzymatic degradation in the body, high gel turbidity, and high water content [12,13,14].

Therefore, there is an effort to prepare new starch derivatives or find non-traditional starches to overcome these disadvantages. In addition, there are several challenges in new industrial materials and the food industry based on customer needs that help the environment and human health. New trendy food products based on plants include snacks and food with “clean-label” starches, which respect the environment by producing no chemical by-products [15] or containing resistant starch. The hypothesis of this paper was to collect enough data from several years ago about the physicochemical properties of non-traditional starches to enhance their value-added application. The reference papers contain a detailed arrangement of the analytical methods and setting of instruments. A novelty of this research is waste utilization for isolation to obtain a new type of starch.

## 2. Isolation of Starch

The industrial isolation of starch depends on many factors. The first step can be the steeping of the grain (e.g., tapioca or corn), cold extraction of the mashed product (e.g., potato), or preparation of dough or slurry from flour (e.g., wheat). Most native starches have a unimodal distribution of granule size. Cereal starches are bimodal, so one needs to separate starch granules into two fractions. The botanical origin, weather conditions and environmental factors, farming techniques, and procedure used for starch isolation can all influence its chemical composition and structure [16]. 

Table 1 schematically summarizes some tested laboratory procedures for starch isolation from various sources. Of course, the laboratory isolation of nonconventional starches can differ from the common industrial ones. 

The mentioned procedures vary greatly. The first part of the isolation follows from the character of the raw material. Some researchers have used sodium hydroxide solution (0.16–2%) or added protease for dissolving proteins from starch granules. Other scientists have applied sodium or potassium metabisulfite, sodium bisulfite, or sodium sulfite to prevent oxidation and microbial activity. In some cases, they have used organic solvents like hexane or ultrasonification.

## 3. Size, Shape, and Crystallinity of Starch Granules

As will be seen below, the size of starch granules affects the properties and use of the given starch, but on the other hand, the separation technique applied is also important. Small-granule starches have unique properties due to a higher water affinity, larger surface-area-to-volume ratio, and low sedimentation coefficient [21,78,79]. 

Table 2 summarizes the sizes and shapes of starch granules depending on their source. The smallest granules (<5 µm) with a high specific area were found in *Agriophyllum squarrosum*, amaranth, foxnut, and gorgon nut seeds and in the leaves of *Arabidopsis thaliana.* Large particles can be suitable for good separation; the diameter D [4,3] is defined in terms of the moment-ratio system as:$$D\left[4,3\right]=\frac{\sum n_{i}D_{i}^{4}}{\sum n_{i}D_{i}^{3}}$$
where *n_i_* is the number of the measured particles and *D_i_* is their diameter. Kidney bean and canistel starches demonstrate a high value of this parameter. Ma et al. [80] measured out 200 µm in oat seeds. Chavez-Salazar et al. [65] found 65 μm long particles in unripe plantains. The largest average size (47 μm) was starch from achira rhizomes. The shape of particles highly varies and can be spherical, oval, kidney, elongated, discoid, conical, polygonal, triangular, or irregular.

Native starches include A-, B-, and C-types according to their crystallinity determined using X-ray diffraction. A- and B-type starch contains only A- and B-type crystallinity, respectively, but C-type starch contains both A- and B-type crystallinities. According to the proportion of A- and B-type crystallinity from high to low, C-type starch is usually divided into C_A_, C_C_, and C_B_-types [81]. Only a few starches possess B-type crystallinity, specifically those from achira rhizomes, lily bulbs, *Phoenix sylvestris*, and ulluco tubers, or pumpkin and winter squash fruit (Table 2).

## 4. Chemical Structure, Microstructure Observation

The length and configuration of the chains in amylopectin influence many properties such as the swelling power, water solubility index, crystallinity, pasting and thermal properties, main gelatinization temperatures [71], retrogradation [33], syneresis, resistant starch formation and digestibility [35,61], hardness and cohesiveness of the gels [42,61], capability for esterification and functionality of the derivative [65], and resistance to enzyme hydrolysis [77].

Hanashiro et al. [93] defined branches of amylopectin as A, B1, B2, and B3 types having degrees of polymerization of 6–12, 13–24, 25–36, and >37, resp. The chain length distributions of starch samples were determined using a high-performance anion-exchange chromatography system [27]. Small granules like those from send rice (*Agriophyllum squarrosum*) or quinoa starches had a high percentage of A-chains and a low percentage of B3. Similarly, the number-average degree of polymerization of amylopectin decreased with a decreasing granule size in barley starch [94]. On the contrary, the minimum number of A-chains was in canna starch and high-amylose corn starch, and *Chlorella*, canna, and tapioca starch with high-amylose corn starch had a maximum B3 chains content (Table 3).

## 5. Data of Differential Scanning Calorimetry, Pasting Properties

Differential scanning calorimetry (DSC) on starch is indicative of interfering with differences in starch structures, changes in the physical states of starch, and the interactions of starch polymers with other constituents and composite food matrices. This method has enabled the measurement of gelatinization temperatures, provided evidence that chemical modification has reduced or eliminated aggregation and the association of starch molecules during the cold storage of pastes, and evaluated the temperature and enthalpy of the formation of amylose–lipid complexes [1].

The temperature at which gelatinization occurs and the heat enthalpy are closely connected to the starch structure and molecular arrangement. Furthermore, the amylose content could also influence the enthalpy of gelatinization. The thermal data found in recent papers are in Table 4. From an industrial point of view, the onset temperature (To), which represents the initiation of the gelatinization stage in starch granules, is associated with the costs of the used processes in terms of energy and time, and thus a low temperature is required by manufacturers. This low To was found in acorn starch [19]. On the other hand, the highest To (142.9 °C) was found in sago starch isolated from pith waste fiber with a high crystallinity structure [69]. The starch melting temperature range (Tc–To) indicates the homogeneity and quality of the amylopectin crystals [96]. A narrow melting range corresponds to amylopectin crystals with a more homogeneous quality and uniform stability and vice versa [97]. The narrow range of (Tc–To) characterizes starch gained by ginger (7.4 °C), sand rice (8.4 °C), winter squash var. Heili (8.5 °C), avocado seeds (9.2–10.7 °C), *Carioca* beans (10.1–13.8 °C), common corn (10.4 °C), and ramon seeds (10.6 °C).

The gelatinization endotherm is associated with the gelatinization of the crystalline amylopectin lamella, but not necessarily to the proportion of the crystalline area, which corresponds to the amylose: amylopectin ratio and the distribution of amylopectin chains in the granule [37]. A higher enthalpy can also indicate a higher content of starch–lipid complexes, which requires more energy to disrupt [80]. Starches from *Euryale ferox* seeds and sago pith waste fiber demonstrated the highest enthalpy.

The retrogradation enthalpy (Δ*H_ret_*) reflects the energy required to melt the recrystallized starch structure and starch molecular double helices formed during long-term storage. There are not enough data for a comparison of Δ*H_ret_*. The highest value was found for sand rice starch (its gel exhibited significant recrystallization), while *Phoenix sylvestris* starch demonstrated the lowest value.

As for the pasting properties, viscosity describes the magnitude of the friction generated by the mutual movement of starch granules after gelatinization [100]. A comparison among the data from the references is very difficult, because the input concentration of the tested suspensions in aluminum vessels differed from 3% (db *w*/*w*) to 16.67% (*w*/*w*)—see Table 5.

The peak viscosity (PV) of starch indicates the maximum swelling caused by heat in the presence of water [101]. When we exclude the values for extremely high or undefined starch concentrations, the highest PV was found in samples of starches coming from ulluco, winter squash, bean, pumpkin, quinoa, and sweet potato. On the other hand, waxy rice and ginger starch had a low PV by a higher starch concentration.

Breakdown (BD) equals PV minus trough (minimum) viscosity [102]. BD represents the thermal stability of starch paste [40] and is the function of the rigidity of the swollen starch granules [103]. The authors found the maximum BD in mung bean, quinoa, and sweet potato starches. The lowest values were in ginger, foxnut, and normal corn starches. A low breakdown viscosity indicates a high paste stability to remain intact at a high temperature. This characteristic of starch will be suitable for food products requiring a high processing temperature, and alterations in viscosity are unwanted during heating and cooling [104].

High final viscosities (FV) were found in Turkish bean and winter squash starch. On the contrary, this was low in waxy amaranth. Setback (SB) equals FV minus trough (minimum) viscosity [103]. SB indicates the recrystallization degree of starch during the cooling of starch paste, especially the recrystallization and rearrangement of amylose molecules [80]. Waxy varieties and chestnuts demonstrated a reduced setback viscosity, while Turkish beans, rice, and quinoa showed the highest SB.

## 6. Swelling Power, Solubility and Water Absorption

The extent of granule swelling is determined by measuring the swelling power (SP; g/g), which is reported as the ratio:$$SP=\frac{m_{g}}{m_{st}-m_{sol}}$$
where *m_g_*— the weight of the gel (g), *m_st_*—the weight of the starch sample (g), and *m_sol_*—the soluble weight (g).

SP measures the hydration capacity, because food-eating quality is often associated with the water retention in the swollen starch granules [105].

Solubility (SW; %) is the percentage of molecules leached from the starch granules after swelling under specific conditions:$$SW=\frac{m_{sol}}{m_{st}}100$$

The SP and SW values provide information about the degree of interaction between the chains present in the crystalline and amorphous regions of the starch granules and are correlated with amylose content [71]. A higher amylose content in starch contributes to a lower functionality, which could limit its application in food-processing industries such as bakeries [24]. In addition, the formation of protein or lipid complexes also influences the SP [64].

The water absorption index (WAI; g/g) is defined as:$$WAI=\frac{m_{g}}{m_{st}}$$
and had similar values to the *SP*. The results of the swelling power solubility and water absorption of different starches, at temperatures ranging from 60 to 80 °C, are presented in Table 6. High values of SP were found for starches of canna, quinoa, annatto, *Agriophyllum squarrosum*, and *Phoenix sylvestris* at 80 °C. All starches showed an increasing swelling power at increasing heating temperatures because of gelatinization.

Starches of canna, quinoa, black bean, and *Phoenix sylvestris* showed the highest SW values measured at 80 °C, while anchote, ginger, and foxnut starches demonstrated the lowest SP values.

## 7. Refrigeration and Freezing Stability of Starches

Frozen foods containing starch can be repeatedly freeze–thawed during storage and transportation. Thermal fluctuations in foods during transport cause water melting and recrystallization, lowering their quality and acceptability. Simultaneously, the starch recrystallizes during cooling, resulting in water separation known as syneresis. Syneresis is an undesirable phenomenon that results in a poor starch gel texture [101].

Chemical modifications such as acetylation and hydroxypropylation can positively influence freeze–thaw stability. These processes have also been tested on non-traditional starches [106].

Zarroug et al. [19] studied the stability under refrigeration and freezing in different native starches during storage times of 24, 48, 72, 96, and 120 h. According to Wu et al. [80], adzuki bean starch had a higher amylose content, swelling capacity, and freeze–thaw stability.

Hussain et al. [101] published the percent syneresis of corn, sweet potato, and Turkish bean starch gels. Vázquez-León et al. [35] dealt with the stability under refrigeration and freeze–thaw of black bean starch gel, measured as a percentage of syneresis (water exuded), determined after the 1st–5th cycles. A comparison of the cited data is not possible due to the different calculations of syneresis and low scope of the references. We can just make the following provisional conclusions:-the lower the temperature of the refrigeration, the higher the syneresis,-the longer the storage time, the higher the syneresis,-the syneresis of acorn starch is comparable to corn starch,-the syneresis of sweet potato starch and Turkish bean starch is lower than that of corn starch.

Photinam et al. [107] reported that freezing at low temperatures after gelatinization resulted in the recrystallization of starch molecules and a higher resistance to enzymatic hydrolysis. Rodboontheng et al. [108] found that canna starch granules subjected to freezing were more intact and smaller compared to those without freezing. Upon freezing, the swollen granules shrank to smaller-sized granules.

According to Liu et al. [44], native chestnut starch has characteristics similar to corn starch, such as a higher swelling power, better freeze–thaw stability, and lower pasting and gelatinization temperatures, which has made it potentially a good source of starch.

## 8. Hydrolysis, Digestibility

As the principal energy source of the human body, the digestibility of starch is an important parameter in the quality assessment of food. During starch digestion, glucose is released. Thus, starch digestion is closely associated with the postprandial blood glucose and insulin levels in humans [84]. It is influenced by various factors, for example, structural characteristics, the formation of starch–protein complexes (containing strong disulfide or tannin bonds [109]), interactions with polyphenolics [110], β-carotene [111] and other components, and preheating treatment. Wu et al. [80] dealt with digestibility of prepared complexes of adzuki bean starch-fatty acids.

The amorphous area of the starch molecule is more amenable to acid attacks, being more easily and quickly hydrolyzed [29]. However, a high amylose content in starch decreases its digestibility. A smooth surface of starch granules, without grooves, hinders the enzyme’s diffusivity inside the granule and delays the hydrolysis [37]. Their size can also affect their digestibility, as the surface area and starch volume affect the interaction between the starch substrate and enzyme. Therefore, the large surface area of small granules has resulted in a high rate of enzymatic hydrolysis [94,112].

The lower digestibility of starch results in lower obesity and microbial formation of short-chain fatty acids in the human gut, thus having healthy effects [110]. Englyst et al. [113] divided starch into rapidly digestible starch (RDS; starch digested from 0–20 min), slowly digestible starch (SDS; digested from 20–120 min), and resistant starch (RS; remaining starch) based on simulated digestion properties in vitro and bioavailability. RS can be used as dietary fiber, reducing calorie intake (low glycemic index), preventing fat deposition and colon cancer, and enhancing mineral absorption [22]. RS includes five groups.

The lower in vitro digestibility of legume starches than that of cereal starch might be due to their high amylose content (because of their lower susceptibility to enzymatic hydrolysis), the absence of surface pores on the granules, large amounts of B type crystallites, strong interactions between amylose chains, intact cell structures enclosing starch granules, and large amounts of soluble dietary fiber [35].

The starch molecules in freshly gelatinized starches are amorphous and susceptible to digestive enzymes [71]. Transglycosylation forming pyrodextrins is responsible for creating new glycosidic bonds, which, in turn, reduce digestibility [29].

Table 7 includes some new results of starch digestibility. Unripe plantain, pumpkin, and winter squash Yinli starches demonstrate the lowest RDS (<3%) and the highest RS (>90%). Indigenous Himalayan folk rice cultivars are good examples of the dependence RS on amylose content. For an amylose content in the range of 0.7–29.1%, RS was low, i.e., 0.4–2.3% [72], but for a higher amylose content of 31.8–40.7%, RS rapidly increased to 85.4–92.8% [73].

However, most of these starches are not eaten in their native form. A low RDS is maintained in *cooked* lily starch (20%) and a high RS content was found in lily starch (>50%) and unripe plantains (>25%). The highest SDS was found in the native starch of annatto and Euryale *ferox* (>35%). A very high SDS content was found in the thermally (80–100 °C) processed starch of gorgon nut.

Besides the evaluation of RDS, SDS, and RS, simulations of starch digestion in the small intestine were performed using various evaluation models [61,87]. The first-order rate equation was estimated for the digestibility kinetics according to the following formula:$$C_{t}=C_{\infty }.(1-e^{-kt})$$
where *C_t_* and *C_∞_* (mg/mL) are the digestibility of starch when the hydrolysis time is *t* (min) from 0–∞, respectively, and *k* indicates the kinetic coefficient of the starch digestibility (min^−1^). Liu et al. [87], Mahajan et al. [61], and Wu et al. [114] evaluated *C_∞_* at 42.6% and *k* at 0.029 min^−1^ for chestnut starch, *C_∞_* at 64.0% and *k* at 0,0264 min^−1^ for Kutki millet starch, and *C_∞_* at 66.5% and *k* at 0.0297 min^−1^ for adzuki bean starch, resp., compared to *C_∞_* at 65.6% and *k* at 0.0332 min^−1^ for corn starch. Mahajan et al. [61] also listed other digestion models, e.g., the Duggleby model, Paolucci-Jeanjean model, and Weibull model.

The simple magnitude is the degree of starch digestion or enzyme hydrolysis (%) defined as:(1)$$DEH=\frac{{100.m}_{HS}}{m_{IS}}$$
where *m_HS_* (g)—the weight of hydrolyzed starch and *m_IS_* (g)—the weight of initial starch [108,115]. The rate of digestion is closely related to the glycemic index.

## 9. FTIR Spectral Analysis

Fourier-transform infrared spectroscopy (FTIR) is a powerful tool for characterizing the structure of starch and starch-based composite films [25,89,116]. This technique enables (i) identifying impurities in isolated starch, (ii) identifying chain conformation, single- and double-helical structures of starch, (iii) evaluating the crystallinity of starch-based products, (iv) identifying functional groups in chemically modified starches, and (v) identifying changes in starch-based biodegradable films.

The observed FTIR spectra are similar for all studied starches [25,117]. A broad and intense band of 3700–3000 cm^−1^ represents the stretching vibrations of O–H bonds in hydroxyl groups and overlap narrow weaker bands at 2928–2931 cm^−1^ attributed to the stretching vibration of the C–H bond in glucopyranosyl residues. The bands observed at 1153, 1086, 1012, and 930 cm^−1^ in the spectral region, often assigned as “sugar region”, are attributed to C–O–C, C–O, and C–C stretching and C–O–H bending vibrations in the glycosidic linkages and pyranoid rings [19,25]. Several bands at 921–930 cm^−1^ and 767 cm^−1^ are characteristic for 1,4-α-glycosidic bonds in starches [30,64], and the band at 860 cm^−1^ arises from C1α–H and CH_2_ deformations and is indicative of the α-anomer of glucopyranose [118]. Finally, several bands at 708 cm^−1^ and below arise from the skeletal modes of the pyranoid ring of α-glucopyranose [119].

Water molecules play a crucial role in starch structure [120] and contribute to the broad absorption at 400–900 cm^−1^ (twisting vibration of bound water), 1640–1650 cm^−1^ (HOH scissoring vibration), and 3500 cm^−1^ (HOH stretching vibration). In hydrated starches, the band near 2100 cm^−1^ can be attributed to the free water content [25,121] and assigned to vibrations from the scissoring and rocking vibrations of the retained water [121]. In addition, hydration causes non-linear changes in the FTIR spectra of starches depending on the contribution of the ordered structure, partially causing a shift in peak position from 1000 to 1200 cm^−1^ [116].

The bands in “sugar region” are sensitive to hydration and the crystalline/amorphous stay of starches. The IR band at 1022 cm^−1^ is related to the amorphous phase [9], while the bands at 993–995 cm^−1^ and 1047 cm^−1^ correspond to the crystalline region of the starch [25,89]. The ratio of the band intensities at 1047 cm^−1^/1022 cm^−1^ and 995 cm^−1^/1022 cm^−1^ quantitatively characterizes the degree of short-range order and the proportion of double helices in starches, respectively [89,116]. For sand rice (*Agriophyllum squarrosum*) and barley starches, the ratio of band intensities at 1047 cm^−1^/1022 cm^−1^ confirmed that polysaccharide chains are rearranged when heated under a low humidity, equal to or below 20%, but at a higher humidity, reaching 30%, short-range ordered structures began to be disrupted [122]. The mentioned ratios can be improved by the addition of another polysaccharide to the starch [84,115,123]. However, the possible overlap of the vibrational bands of these polysaccharides in the “sugar region” may influence the obtained ratios.

The intensities of the bands at 1200–1400 cm^−1^ and at 2940 cm^−1^ increased after the partial acidic hydrolysis of achira (*Canna edulis*) starch [9] and the treatment of pea starch with pullulanase [89]. These bands included CH_2_ vibrations and were sensitive to hydrogen bonds involving starch hydroxylic groups. The increase in these bands may indicate an enhanced number of free CH_2_OH groups due to the hydrolysis of 1,6-α-glycosidic bonds (unbranching).

The FTIR spectra of chemically oxidized cassava starch showed a carboxylate antisymmetric stretching band at 1600 cm^−1^ and some disturbances at 1060 cm^−1^, probably due to stretching of the C1–O5 bond in the pyranoid ring [124]. These spectral changes were consistent with the structural features of this starch suitable for baking properties. The FTIR spectrum of oxidized jackfruit seed starch demonstrated a new band at 1760 cm^−1^ assigned to the C=O stretching vibration of the carbonyl group [54]. The oxidation of lotus root starch led to the appearance of new IR bands at 1610, 1726, and 2820 cm^−1^ due to vibrations of new carbonyl and carboxylate groups [9]. The differences in the FTIR spectra of oxidized lily starches highlighted the sensitivity of crystalline and amorphous structures to specific treatments using the ratio of band intensities at 1047 cm^−1^/1022 cm^−1^ [56]. The FTIR spectra of starches esterified with fatty acids had a new IR band of C=O stretching vibration at 1749 cm^−1^ [125]. Two IR bands of CH_2_ stretching vibrations at 2860 and 2926 cm^−1^ also appeared and increased with the length of acyls. A new IR band of phosphorylated starches at 1417 cm^−1^ corresponded to the P=O stretching vibration [88].

The FTIR spectra of adzuki bean starch–fatty acid complexes demonstrated two narrow bands at 2850 cm^−1^ and 2917 cm^−1^ assigned to the antisymmetric and symmetric stretching of the CH_2_ groups in the long-chain fatty acids. In addition, a weak band at 1383 cm^−1^ assigned to the CH_3_ bending vibration was also found [112]. Additionally, the bands at 1635–1656 cm^−1^ (amide I), 1545 cm^−1^ (amide II), and 1240–1350 cm^−1^ (amide III) are attributive to the vibrations of amide groups in proteins, which are often present in starch preparations [19,25].

Composite starch-based edible films often contain proteins, lipids, aromatics, and other polysaccharides. FTIR spectroscopy is able to detect the presence of these compounds via characteristic vibration bands. Starch-based film represents specific bands at 3326 cm^−1^, corresponding to O–H stretching in glycerol, starch, and water [62]. The FTIR spectra of biodegradable edible film obtained from arrowroot starch and iota-carrageenan showed characteristic bands of ι-carrageenan at 1249–1269, 924–926, and 842–849 cm^−1^ assigned to S=O stretching, 3,6-anhydrogalactose, and galactose-4-sulfate vibrations, respectively [3]. Hussain et al. [101] observed two IR bands at 2927 and 2940 cm^−1^, indicating the asymmetric stretching vibration of the CH_3_ groups of the acetyl and methyl esters in cactus (*Opuntia ficus-indica*) and acacia (*Acacia seyal*) gums in a mixture with various starches. The IR band observed at 1721–1728 cm^−1^ indicates the C=O stretching vibration of the carboxylic acids and esters in polysaccharidic gums. The bands at 1415–1400 and 1310–1230 cm^−1^ corresponded to the symmetric stretching of the COO^-^ of the carboxylate groups and C–O stretching of the carboxylic acids in the gums, respectively.

## 10. Raman and NMR Spectroscopies

In addition to FTIR, Raman spectroscopy is also widely used in the structural analyses of starches, for example in the characterization of the structural changes in retrograded [126,127], microwave-treated [119], and irradiated [118] starches. FTIR and Raman spectra were used in the analysis of biodegradable starch-based films containing saturated fatty acids [128]. Both of these vibrational spectroscopy techniques were applied to evaluate how heat treatments and extraction procedures affect the properties of ajocotte starches [129].

The Raman spectra of starches are sensitive to the presence and type of crystalline structure, degree of substitution, and relative amounts of amylose and amylopectin. The band intensity in the Raman spectra for starches depends on their structural features [130]. The typical Raman bands of starches near 2910, 1458, 1260, 1124, 864, and 475 cm^−1^ correspond to C–H stretching, C–H, CH2, and C–O–H bending, C–O–H, C–C–H, and O–C–H bending, C–O and C–C stretching, C–C–H and C–O–C bending, and pyranoid ring skeletal vibrations, respectively. In amorphous starch, Raman bands, except those at 2910 and 478 cm^−1^, were weak and almost disappeared at low frequencies. The Raman intensity varies directly with the size of the crystalline region in starch. The Raman bands for B-type starch (small crystalline part) were less intense than those for A-type starch (large crystalline part), and the most of these bands shifted to lower wavenumbers. De Gussem et al. [131] observed evident shifts in the Raman bands of amylose at 405, 481, 757, and 855 cm^−1^ compared to the corresponding bands of amylopectin at 410, 477, 769, and 865 cm^−1^. The skeletal vibration band of starch near 477 cm^−1^ is one of the most intense in the Raman spectrum and can be used as a marker to detect starch in mixtures with other polysaccharides [132].

Since Raman spectroscopy is less sensitive to the contribution of water and polar groups, it is more prospective in the analysis of aliphatic moieties in substituted starches. For example, no significant changes were observed for cationic quaternary ammonium starches in the FTIR spectra due to the overlap of the strong OH and H_2_O stretching bands [133]. In contrast, the corresponding Raman spectra showed several bands at about 3030, 970, and 761 cm^−1^ assigned to the vibrations of trimethylammonium substituents. The ratio of the band intensities at 761 cm^−1^/938 cm^−1^ (reference starch band) can quantify the degree of substitution. Raman spectroscopy was applied to determine the degree of succinylation in modified starches [134].

Nuclear magnetic resonance (NMR) spectroscopy is a powerful tool in the structural analysis of natural and modified polysaccharides, including starches. Applications of this method in the characterization of starch systems were reviewed by Zhu [135]. Various techniques of NMR spectroscopy can (i) characterize the structure and physical properties of native and modified starches, including the effects of moisture and the impact of physical treatments, (ii) analyze starches in gel and solid states, (iii) identify enzyme susceptibility and resistant starches, (iv) identify and quantify functional groups in chemically modified starches and starch-grafted copolymers, and (v) analyze the interaction of starch with non-starch components in food and other complex systems. For example, ^1^H NMR has been applied to the determination of the degree of substitution of acetylated and other acylated starches [136,137,138], and octenyl succinic anhydride-modified waxy corn starch was characterized using ^1^H and ^13^C NMR [139].

Solid-state ^13^C CP/MAS NMR spectra can evaluate the molecular organization of starch samples [5,63,89]. Broad resonance signals at 94–105 and 80–84 ppm correspond to the C1 and C4 carbons, respectively, involved in 1,4-α-glycosidic bonds. The former resonance signal is sensitive to the crystalline structure of starch and can split into 2–3 components, while the latter resonance signal indicates the amorphous structure in starch. For example, oat starch had a triplet peak at 101.36, 99.23, and 98.84 ppm, typical of an A-type crystalline structure [5]. After treatment at 500 MPa for 15 min, this region transformed into a single strong peak at 101.36 ppm, and the relative area of the C4 signal increased, confirming the gelatinization of oat starch. Large and overlapping signals at 68–78 ppm arose from the remaining endocyclic carbon atoms C2, C3, and C5. A signal at 58–65 ppm corresponded to the C-6 carbon atoms of exocyclic CH_2_OH groups [140]. For native and partially degraded pea starch, the ratio of the integral area at 58–65 ppm to the sum of areas of all the carbon signals quantify the relative proportion of double helices in starch, and the ratio of the integral area of the C4 signal to the sum of areas of all the carbon signals quantify the amount of amorphous phase [89]. Alternatively, the total ^13^C CP/MAS NMR spectra deconvoluted by subtracting the scaled NMR spectrum of the amorphous starch to zero intensity at 84 ppm yield the ordered and amorphous phases of the starch samples [122,141,142]. Li et al. [63] studied the cold water swelling of oat starch using subcritical ethanol–water treatment using X-ray diffraction, FTIR, and ^13^C CP/MAS NMR spectroscopy. The obtained results confirmed that the structure of the modified starch changed from A-type double helical to V-type single helical and the short-range crystalline structure decreased as well.

Time domain ^1^H NMR can evaluate the molecular mobility and water distribution in the starch–water systems [143]. This method can be applied to elucidate the molecular dynamics of starch transitions during gelatinization in dough/batter systems during heating/cooling and used in the quality control of starchy food products.

## 11. Starch Modification

Native starch may be modified in numerous ways, for example, physically, chemically, or via an enzyme action.

Pregelatinized starch originates when starch slurry is heated by steam [86]. It is a common ingredient in foods, as well as other industrial applications. The other tested physical methods for starch modification are microwave treatment [31], annealing (ANN), heat moisture treatment (HMT), high-pressure homogenization [5,43], ultrasound [22,69], or ultrasound combined with plasma treatment [40].

Biduski et al. [144] found that HMT increased the peak viscosity and breakdown while reducing the final viscosity and setback of the *Pinhão* starch. There was an increase in the gelatinization temperature and a decrease in the ΔH after treatment. HMT changed the short-range ordered structures, shifted the X-ray pattern from C-type to A-type, and reduced the relative crystallinity. Akinyosoye and Nwokocha [76] investigated the effect of HMT on the properties of *Treculia africana* starch. As the moisture level of HMT increased, the swelling decreased, and there was an increase in amylose leaching. Pasting temperature and retrogradation increased, and breakdown decreased with HMT. Thus, HMT strengthened the granule structure, resulting in starch which can withstand severe processing conditions without viscosity loss. Similarly, Wu et al. [74] tested the effect of heat-moisture treatment on the structural and physicochemical characteristics of sand rice starch. According to Rodboontheng et al. [108], swollen starches prepared via HMT are promising encapsulating agents for food ingredients and supplementary foods such as essential oils, nutritious oils, flavors, and pre/probiotics.

The ANN treatment applied to ulluco starch proved to be an efficient method for improving the physicochemical parameters such as solubility, water retention capacity, and the texture of the gel formed, which are valuable properties [91].

Natural starch can be modified in various ways to produce acid- or enzyme-modified, oxidized, cross-linked, esterified, or etherified starch, converted into cationic derivatives, or subjected to pyroconversion [29]. These modifications, present at a very low degree of substitution in starch preparations, produce dramatic differences in the physical and chemical properties of starch. The rate of chemical reaction depends on the starch crystallinity and is linked with the porosity and size of starch granules, temperature, pH, and the concentration of the chemical agent.

Partial hydrolysis is a way for the fabrication of acid- or enzyme-modified starches. This process is suitable for the preparation of porous starch, e.g., from banana [28] or lotus [89] starches.

Tung et al. [54] used a hydrogen peroxide agent to oxidize jackfruit seed starch. Xu et al. [9] studied the optimized process conditions of lotus root starches modified via treatment with sodium hypochlorite. Hu et al. [34] dealt with the oxidation of Tartary buckwheat starch by ozone. The product retained a relatively ordered crystal structure, lower amylose content, SP, solubility, digestibility, higher viscosity, and good viscoelastic gels than the native one, resulting from the cross-linking reaction between the aldehyde group produced by oxidation and the hydroxyl group of the adjacent starch in the amorphous region.

Phosphorylation and acetylation change the functional and physicochemical properties of starches. The hydroxypropylation of non-traditional starches enhances swelling capacity, viscosity, clarity, and freeze–thaw stability [86,106].

Also, a combination of physical and chemical modification was tested, e.g., ozone oxidation together with high-pressure homogenization [56] or high hydrostatic pressure and cationization [85].

## 12. Potential Application of Non-Traditional Starches

Generally, starch can be used as an additive, gelling agent, thickener, fat replacer, and emulsifier in crunchy foods, baby foods, and controlled release systems, and as a flavor enhancer and coating agent in many industrial foods, feeds, and drugs to improve their functional and sensory characteristics [19,24,98,112,145]. In the food industry, starch is used mainly to improve the properties of baking flour and bakery products such as cakes, bread, and noodles due to their high consumption. During their formulations, starch is one of the components responsible for broad technological functionality [19]. The push for plant-based food is expected to grow in new categories, e.g., snacks.

The development of new starch raw materials can be used for developing healthy foods required by consumers, thereby enhancing the value-added application [52]. They are increasingly choosing labels that help choices about the environment and human health. So-called “clean-label” starches represent a relatively new group, not marked with the E index in food (they are not chemically modified). Their advantage is that they are environmentally responsible by not producing chemical by-products [15]. Starches from new sources could replace chemically modified starches due to their properties being similar to commercial modified ones.

Resistant starch (not accessible to human digestive enzymes) has beneficial effects for health; it positively influences the digestive tract and its fermentation in the colon causes the formation of short-chain fatty acids and the lowering of the pH. It further increases fecal bulk, protects against colonic cancer, improves blood cholesterol levels, assists glucose tolerance control in diabetes, and causes lower blood lipid levels. However, the current RS/dietary fiber intake by Europeans is insufficient. Therefore, it is suitable to add it to food. According to Du et al. [146] and Yuan et al. [66], native beans, winter squash, and pumpkin starches are the RS sources in formulations with desired fiber-like benefits like a lower digestibility.

Some papers have solved the utilization of starch from wastes, e.g., coming from seeds of annatto [25], avocado, *Chenopodium*, and jackfruit [42,54,83,98]. Besides being used in the form of native starch from roots, tubers, seeds, and wastes, modifying non-conventional starches could provide options for extending the required engineering properties [54]. Octenyl-succinic-modified starches from various sources have been widely used to encapsulate food ingredients by forming emulsions. Moreover, nonenyl succinic quinoa-modified starch could be used for the fabrication of emulsions [99].

In recent years, the importance of high-amylose starch has grown due to its expanded range of functionalities compared to native starch. These functionalities include nutrition, food processing, medicine, and industrial use [33].

An outstanding group of starches covers starches having small granules. Cow cockle starch (0.3–1.5 μm), rice starch (2–8 μm), and amaranth starch are suitable as fat substitutes in frozen dessert and mayonnaise; similarly, acorn starch has been used in custard because the small particle starches could form a smooth creamy fat-mimetic texture [79,145,147]. Wheat B-starch with a particle size fraction below 8–10 µm has found uses in the production of alcohol, dextrin, and feed, in a mixture with bentonite for foundries [148], for hydrolysates, and for energy use in biogas units [15]. It has also been tested for use in biodegradable plastics [149]. The advantage of starches having small granules is their high specific surface.

Nanomaterials exhibit properties that are distinctive and qualitatively different from large-size particles. Starch nanoparticles (SNPs) developed from mango starch, mungbean, and water chestnut starch are comparable with wheat or potato nanoparticles. SNPs also have great potential for papermaking use, surface sizing, coating, and in paperboard as a biodegradable adhesive instead of petroleum-based adhesives. All the tested SNPs are non-cytotoxic and can be safely used in biomedical applications [150]. Nanocomposites produced with mung bean starch showed the best properties among those with different starches used as matrices. The higher amylopectin ratios in starch result in the better interaction, dispersion, and distribution of nanofillers in starch-based nanocomposites due to the branched molecular structure of amylopectin. Sago starch nanocomposites produced with zinc oxide nanorods increased the electrical conductivity [151].

Plastic-based synthetic polymers widely used in modern food packaging contribute to plastic-waste end-products, which accumulate in and pollute terrestrial and aquatic environments. Thus, there are ways of reducing the use of and dependence on them [3]. Low cost, low density, decreased tool wear, renewability, and degradability [152] are all recognized benefits of starch resources for the industry manufacturing biodegradable plastics. Involving starch in bioplastics provides in two ways, i.e., in the form of a filler with intact starch granules or incorporation directly into the composite matrix, which is so-called thermoplastic starch [153]. Waste or by-products from sugar palms, cassava, or corn are suitable as polymer composite reinforcements [154]. On the other hand, poor resistance to water and low strength are limiting factors for materials manufactured from starch [155]. Blending starch with other polymers such is PVA is an effective and convenient way of overcoming the shortcomings of starch materials [45]. The use of nanotitanium dioxide nanoparticles and essential cinnamon oil incorporated into sago starch-based bionanocomposite films show excellent antimicrobial activity against *Escherichia coli*, *Salmonella typhimurium*, and *Staphylococcus aureus*, which results in a high potential for use in active packaging in food industries. Nanocomposite films processed with sugar palm starch, sugar palm nanocrystalline cellulose (SPNCC) with cinnamon essential oil have similar effect [151].

Edible biodegradable films (EF), intended for primary food packaging use, are made from food-grade and biodegradable polymers. Therefore, edible films are used to wrap the food product to inhibit the moisture uptake, extend the shelf life, and are consumed together with the food. Consequently, EF can generate almost zero waste after disposal [3]. EFs fabricated from starch polysaccharides are one of the primary genuine forms of biopolymer materials for food packaging due to strength, processing, product adaptability, rheological properties, and performance. Additionally, these products are environmentally friendly and cost-effective. EF development from sesame seed gum, mung bean, cassava, and avocado seed starches has been widely studied [23]. Also, native quinoa, amaranth, and ulluco starch are ideal materials for the fabrication of EF with a good mechanical strength, thermal stability, and antimicrobial activity compared to films made from corn starch and yam starch [77,156,157]. Materials with a low opacity, like EF, can be used in food packaging for improved product visualization and consumer acceptance [144].

Several non-conventional starches have technological advantages over common starches for specific applications in the cosmetic and paper industries [145]. In addition, starch can be applied as an adsorption material in many branches [15,28,108]. The biotechnological application of duckweed starch instead of commonly used starch feedstocks via the highly efficient fermentation of glycerol was tested by Yang et al. [48].

## 13. Conclusions

Nowadays, the research focuses on novel starch sources in the context of food security, sustainable production, the use of by-products, e.g., coming from annatto and avocado, their regional availability, and technological advantages over traditional starches.

Published numerous data like shape, size, and crystallinity characterize non-traditional starches. Small-granule starches demonstrate unique properties due to a higher water affinity, larger surface-area-to-volume ratio, and low sedimentation coefficient. Because of their smooth texture, they can be utilized as fat substitutes in frozen desserts, mayonnaise, and custard. These starches have shown an excellent compatibility in biodegradable plastics. They also can be applied in their native or modified form as adsorbents.

The practical use of starches in the food industry requires knowledge of the thermal and pasting properties of the suggested starches. The low onset temperature (To) in acorn starch is associated with the costs of the industrial processes in terms of energy and time. The requested high retrogradation enthalpy was characteristic of sand rice starch, the smallest one for starch from *Phoenix sylvestris* with a high swelling power.

Starch plays a crucial role in the food industry as a thickening agent. Starches from ulluco, winter squash, bean, pumpkin, quinoa, and sweet potato demonstrate a high peak viscosity (PV); on the other hand, waxy rice and ginger starch had a low PV. Low setback (SB), indicating recrystallization during the cooling of starch paste, was found in waxy varieties and chestnuts. Turkish bean, sand rice, and quinoa showed the highest SB.

New food trendy categories comprise plant-based food e.g., snacks and food containing “clean-label” or resistant starch. Native, “clean-label” starches from new sources could replace chemically modified starches, due to their similar properties compared to common commercial modified ones. Furthermore, their production does not produce chemical by-products. These techniques for preparing “clean-label” starches include pregelatinization, microwave treatment, annealing, heat moisture treatment, high-pressure homogenization, ultrasound, or ultrasound combined with plasma treatment.

As fiber, resistant starch (RS) has beneficial effects on health. It positively influences the digestive tract and its fermentation in the colon protects against colonic cancer. RS improves blood cholesterol levels, assists glucose blood level control in diabetes, and causes lower blood lipid levels. A high RS content was found in gelatinized lily starch (>50%) and unripe plantains (>25%), and starch from indigenous Himalayan folk rice has a amylose content higher than 32%. The low content of rapidly digestible starch is maintained in cooked lily starch (20%). Thermally processed starch from gorgon nut showed a very high SDS. Therefore, these starches are suitable for a healthy diet.

However, chemical modifications of non-traditional starches are still being tested, including oxidation, phosphorylation, acetylation, hydroxypropylation, and a combination of physical and chemical approaches. Spectroscopy methods are powerful tools for the characterization of the purity, structure, and physical properties of starches, their derivatives, and composite materials based on them. In addition to X-ray diffraction, FTIR, Raman, and NMR can evaluate the ratio between the crystalline and amorphous forms of starch.

Native and modified non-traditional starches can be used also outside the food industry. Materials from sugar palm, cassava, corn, and others have been discovered to be suitable as polymer composite reinforcements. Edible foils fabricated from starch polysaccharides are one of the primary genuine forms of biopolymer materials for food packaging due to their strength, processing, product adaptability, rheological properties, and performance; they are environmentally friendly and cost-effective. A relatively new progressive modification of starches enables the creation of nanomaterials, including nanoparticles/nanofillers, nanofibers, or nanocomposites, providing perspectives for various fields of industry.

## Figures and Tables

**Table 1 foods-12-03794-t001:** Laboratory isolation of non-traditional starches.

Plant	Step 1	Step 2	Reference
Achira (*Canna edulis* Ker.)/rhizomes; sweet potato/tubers	Cutting, extraction in water	Sieving, washing	[17,18]
Acorn (*Quercus suber* L.)/kernels	Acorn flour	Water and alkaline extraction	[19]
Acorn/kernels	Shelling, drying, grinding	Soaking in n-hexane 6 h, in ethanol 30 min, centrifugation, washing	[20]
Amaranth/seeds; oat/seeds	Soaking water 24 h	Sieving, centrifugation	[21]
Amaranth/grains	Soaking 87.5 mM NaOH, washing, grinding	Sieving, centrifugation	[22]
Anchote/seeds	Grinding	Extraction 0.075% (*w*/*v*) Na_2_S_2_O_5_	[23]
Anchote/tubers	Peeling, slicing	Drying, grinding	[24]
Annatto seeds	Depigmented with KOH	Drying, grinding	[25]
*Arabidopsis thaliana*/leaves	Freezing, grinding	Extraction pH 7.4 + EDTA, centrifugation	[26]
Avocado/fruit	Peeling, cutting	Grinding, pasting	[27]
Banana/fruit; breadfruit (*Artocarpus altilis*)	Peeling, cutting	Macerated, sieving	[28,29]
Banana/fruit	Extraction in 4% (*w*/*v*) NaHSO_3_ 4 h	Centrifugation, drying	[30]
Barley/hull-less kernels	Microwave oven, extraction in 0.1 M NaOH 12 h	Washing, grinding	[31]
Bambara groundnut/germinated seeds	Steeping in water 6 h, hydration-germination 24, 48, 72 h, grinding	Sieving, extraction in 0.3% (*w*/*v*) NaOH 4 h, decantation, sieving	[32]
Bean (*Phaseolus vulgaris* L.)/seeds; *Tartary* buckwheat/flour	Extraction in 0.15 or 0.20% NaOH 16 h	Sieving, washing	[33,34]
Black bean/seeds	Extraction in water 12 h	Grinding, sieving	[35]
Canistel/seeds	Drying, dehulled, grinding	Addition of n-hexane, washing, centrifugation	[36]
*Carioca* beans	Extraction in 0.16% (*w*/*v*) Na_2_S_2_O_5_ 8 h	Sieving, centrifugation	[37]
Mung bean/seeds; mango/flour	Soaking 6 h, mixing	Sieving, washing	[38,39]
Mung bean/seeds	Soaking 4 h, mixing	Sieving, washing	[40]
Mung bean/seeds	Wet milling	Washing 0.2% NaOH, centrifugation	[41]
*Chenopodium album*/grains	Drying at 40 °C 24 h, grinding	Extraction 0.25% (*w*/*v*) NaOH 24 h, wet grinding, filtration	[42]
Chinese chestnut (*Castanea mollissima* BL.)/kernels	Peeling, wet-milling	Filtration, decantation	[43,44]
Water chestnut (*Trapanatans* L. var. bispinosa)	Drying 24 h, grinding, extraction in 0.2% NaOH, pH 9	Filtration, decantation	[45]
*Chlorella* sp. MBFJNU-17; quinoa/seeds	Steeping in 0.45% (*w*/*v*) Na_2_S_2_O_5_ 12 h	Grinding, sieving, centrifugation	[46]
*Chrysophyllum albidum* (African Star Apple)/kernels	Soaking in n-hexane 72 h, extraction in 0.05 M NaOH, pH 8	Centrifugation, washing	[47]
Duckweed (*Landoltia punctata*)	Wet milling 0.1% NaOH, incubation 4 h	Decantation, neutralization, washing	[48]
Elephant foot yam starch (*Amorphophallus paeoniifolius*)/tubers	Peeling, milling,	Washing, filtration	[49]
*Euryale ferox*/seeds	Milling, extraction in water 5 h	Sieving, centrifugation	[50]
Foxnut (*Euryale ferox* Salisb.)/kernels	Drying 60 °C, 24 h, extraction in water 5 h,	Filtration, washing	[16]
Gembili (*Dioscorea esculenta* L.)/tubers	Peeling, cutting, mixing	Decantation	[51]
Ginger (*Zingiber officinale* Roscoe.) starch/waste product	Dispersed in water, stirring	Washing	[52]
Gorgon nut/seed	Extraction in 0.20% Na_2_SO_3_ 5 h, washing	Crushing, filtration, centrifugation	[53]
Jackfruit (*Artocarpus heterophyllus* Lam.)/seed	Soaking 2% NaOH 0.5 h, peeling,	Wet milling 0.1% NaHSO_3_, filtration, decantation	[54]
Jackfruit/seeds	Soaking water (6 h and 8 h), stirring	Filtration, washing	[55]
Lily (*Lilium* spp.)/bulb	Homogenization	Filtration, washing	[56]
Lotus/seed	Homogenization with water	Filtration, washing	[57]
Lotus/stem	Cutting, extraction in 0.12% K_2_S_2_O_5_ + 0.25% citric acid, 1 h	Grinding, filtration, washing	[58]
Unripe mango (*Mangifera indica* L.) of Haden and Palmer cultivars/pulp	Cutting, drying, milling	Sieving, washing	[59]
Mango/kernels	Wet milling	Sieving, washing	[60]
Dehusked Kutki millet/grains	Grinding, extraction in 0.25% NaOH 24 h	Sieving, washing, centrifugation	[61]
Proso millet (*Panicum miliaceum* L.)/grains	Extraction in 0.15% Na_2_S_2_O_5_ 15 h	Washing, grinding, sieving	[62]
Oat/seeds	Mechanically flaked, defatted, grinding	Incubation in water + protease 24 h, 40 °C, centrifugation	[63]
*Phoenix sylvestris*/root tuber	Cutting, wet grinding	Filtration, decantation, washing	[64]
Unripe plantains (*Musa* sp. AAB sub-group Plantain cv. Dominico Harton)	Peeling, cutting, extraction in 0.30% (*w*/*v*) Na_2_S_2_O_5_	Mixing, sieving, washing	[65]
Pumpkin (*Cucurbita moschata* Duch. ex Poir.)/fruit; winter squash (*Cucurbita maxima* Duch.)/fruit	Cutting, mixing	Sieving, centrifugation	[66]
Quinoa/seeds	Soaking water 24 h	Mixing with 62.5 mM NaOH, centrifugation	[21]
Quinoa/seeds	Freezing by liquid nitrogen, grinding, mixing with 12.5 mM borate buffer (pH 10) + 0.5% SDS + 0.5% Na_2_S_2_O_5_	Centrifugation, washing, filtration	[67]
Ramon (*Brosimum alicastrum*)/seeds	Soaking water 24 h, mixing	Filtration, centrifugation	[68]
Ramon (*Brosimum alicastrum*)/seeds	Soaking 1% (*w*/*w*) NaOH 24 h, washing in 0.1% (*w*/*v*) NaOH	Neutralization, centrifuging, washing	[68]
Sago pith waste fiber	Ultrasonication	Filtration, washing	[69]
Sago-kithul palm (*Caryota urens*)/flour	Steeping in 0.50 M NaOH 30 min, centrifugation	Washing, neutralization, filtration	[70]
Sand rice (*Agriophyllum* *squarrosum*)/seeds	Dehulled seeds	Alkaline extraction 12 h	[71,72,73,74]
Talipot palm (*Corypha umbraculifera* L.)/flour	Alkali (0.05 M NaOH) method	?	[75]
*Treculia africana*/seeds	Milling, dispersion in 0.2% NaOH	Sieving, washing	[76]
Ulluco (*Ullucus tuberosus Caldas*) tubers	Cutting, homogenization in water	Filtration, washing	[77]

**Table 2 foods-12-03794-t002:** Shape, size, and crystallinity of starch granules.

Starch from	Shape	Size	Crystallinity Type	Reflections of 2θ	Reference
Acorn (*Quercus suber* L.)/kernels	Spherical, oval	5–12 µm	C	15.3° and 23°, 17.4° and 19.4°	[19]
Acorn/kernels	Unimodal mode	Median 8.4 μm	A	15°, 17°, 23°	[20]
Adzuki beans (*Vigna angularis* L.)	Spherical, oval	13–80 μm, average 40.8 μm	C	-	[81]
Achira	Smooth oval or elliptical	-	B	15°, 17°, 20°, 22°, 24°	[9]
Achira (*Canna edulis* Ker.)/rhizomes	Oval, disc	10–110 μm (average size 47.1 μm)	B	5.6°, 17°, 22°, 24°	[17]
Amaranth/seeds	Polygonal with sharp edges	1–5 µm, d_50_ 1.7 µm	A	15.1°, 17.2°, 18.1°, 23.2°	[21]
Amaranth/seeds	-	1–4 µm	-	-	[80]
Amaranth/seeds	-	Bimodal, 1–5 µm; d_50_ 1.7 µm	-	-	[27]
Amaranth/grains	-	-	A	15.5°, 17°, 18.2°, 23.2°, 20.4°, 27.1°.	[22]
Annatto/seeds	Oval (predominant), spherical, triangular, and irregular	Bimodal, 13–197 µm, D_3,2_ 27–31 µm	A	15.1°, 17–18°, 19.9°, 22.7°	[25]
*Arabidopsis thaliana*/leaves	Discoid	0.8–5.0 µm	-	-	[26]
Arrowroot starch	-	-	A	15.15°, 17°, 17.95°	[3]
Arrowroot starch	-	22 × 27 µm	-	-	[4]
Arrowroot starch	Round and polygonal particles, with jagged edges	10–20 µm	A	15.5°, 17°, 18°, 23°	[82]
Avocado/fruit	Lenticular, spherical, irregular	5–30 µm × 4–18 µm	C		[83]
Bambara groundnut/germinated seeds	Smooth oval shaped, some granules irregular and kidney shaped	7–38 µm, average 18–24 µm	A	15°, 17°, 18°, 24°	[32]
Bean (*Phaseolus vulgaris* L.)/seeds	-	D [4,3] 23.8–32.3 µm	-	-	[33]
Black bean/seeds	Polyhedral, rounded	Bimodal, peaks at 6.3 µm and 1 µm	A	15°, 17°, 23°	[35]
*Carioca* beans/seeds	Oval	-	-	-	[37]
Kidney bean	-	D [4,3] 56.2 µm	-	-	[84]
Mung bean/seeds	Spherical, oval, kidney	-	A	-	[40]
Mung bean/seeds	Kidney, round	10–27 µm	C	15.20°, 17.21°, 17.92°, 23.20°	[41]
Mung bean/starch	-	-	C	15.3°, 17.3°, 23.5°	[85]
Red bean/seeds	-	-	A	15°, 16.89°, 17.74°, 19.29°, 22.45°	[86]
Breadfruit (*Artocarpus altilis*)	Spherical	5.7–8.5 μm	-	-	[29]
*Tartary* buckwheat/flour	Most polygonal, some spherical and irregular	-	A	15.1°, 17.2°, 18.0°, 23.0°	[34]
Canistel/seeds		D [4,3] 69.1 µm, d50 38.4 µm	A	15.1°, 17.0°, 18.0°, 23.0°	[36]
*Chenopodium album*/grains	Polygonal, angular	0.6–182.4 μm, median 13.2 μm	-	-	[42]
Chestnut (*Castanea mollissima* BL.)/kernels	Round	-	C	5.6°, 15.1°, 17.0°, 23.0°	[43]
Chestnut	-	-	C	5°, 15°, 17°, 23°	[87]
Chinese chestnut (*Castanea mollissima* BL.) var. Yanshan Zaofeng/fruit	Round, irregular, triangula	-	C	5.6°, 15.1°, 17.0°, 23.0°	[88]
*Chlorella* sp. MBFJNU-17		0.4–1.3 μm, average 1.0 μm	A	15°, 17°, 18°, 23°	[46]
Duckweed (*Landoltia punctata*)	-	1–10 μm, average 5 μm	-	-	[48]
*Euryale ferox*/seeds	Irregular polyhedron	0.5–5.6 μm, mean 2.3 μm	A	15.4°, 17.4°, 18.5°, 23.3°	[50]
Foxnut (*Euryale ferox* Salisb.)/kernels	Polyhedral, angular, oval	1–3 μm	A	15°, 23°, 17°, 18°	[16]
Ginger (*Zingiber officinale* Roscoe.) starch/waste product	Oblong shape with a smooth surface	Average 14 μm	C	10°, 11°, 15°, 17°, 18°, 23°	[52]
Gorgon nut/seed	Regular polyhedrons	1–3 μm	A	15.4°, 17.2°, 18.2°, 23.3°	[53]
Jackfruit (*Artocarpus heterophyllus* Lam.)/seed	-	-	A	15.4°, 17.1°, 18.2°, 23.3°	[54]
Jackfruit/seeds	Round, bell	4.2–13 μm	A	15°, 17°, 23°	[55]
Lily (*Lilium* spp.)/bulb	Round and elliptical	Average 20.5 µm	B	17.0°, 19.5°, 22.2°, 23.6°,	[56]
Lotus/seed	-	7.4–18.3 µm	C	15.10°, 17.10°, 18.00°, 23.10°	[89]
Unripe mango (*Mangifera indica* L.) of Haden and Palmer cultivars/pulp	Bimodal	Main peaks 5 μm and 17 μm	C	5°, 11°, 15°, 17°, 18°, 23°	[59]
Mango/kernels	Oval, spherical, elliptical and oblong		A	14.6°, 23.4°, 18.1°	[60]
Dehusked Kutki millet/grains	Large polygonal and spherical	Median 8.1 µm	A	15°, 17° 18°, 23°	[61]
Proso millet (*Panicum miliaceum* L.)/grains	-	-	A	15.2°, 17.4°, 18.1°, 20.2°, 23.0°	[62]
Oat/seeds		Bimodal, 3–25 µm, d_50_ 8.6 µm	-	-	[27]
Oat/seeds	Oval, irregular	5 µm	A	15.1°, 17.2°, 17.6°, 20.1°, 22.9	[63]
Oat/seeds		Bimodal, 1–200 µm	-	-	[80]
Oat/seeds			A	15.0°, 17.2°, 17.9°, 19.8°, 23.2	[5]
*Phoenix sylvestris*/root tubers	Oval, spherical, flower-shaped-	1 to 10 µm	B	15.09°, 17.36°, 23.32°	[64]
Unripe plantains (*Musa* sp. AAB sub-group Plantain cv. Dominico Harton)	Elongated	20–30 μm wide, 50–65 μm long,	C_B_	5.4°, 15°, 17.1°, 20.4°, 23.1°	[65]
Pumpkin (*Cucurbita moschata* Duch. ex Poir.)/fruit	Dome-shaped	D [4,3] 22 μm	B	5.6°, 15°, 17.2°, 19.8°, 22.3°, 24.0°	[66]
Quinoa/seeds	Polyhedral and angular	1–2 µm, D [4,3] 1.3 µm	A	15.8°, 17.1°, 18.3°, 23.3°	[71]
Quinoa/seeds	Polygonal with sharp edges	1–36 µm, d_50_ 3 µm	A	15.1°, 17.2°, 18.1°, 23.2°	[21]
Quinoa/seeds		0.5–1.2 µm average 1.1 µm	A	15°, 17°, 18°, 23°	[46]
Quinoa/seeds	-	Bimodal, 1–40 µm, d_50_ 4.5 µm	-	-	[27]
Quinoa/seeds	-	Monomodal 0.7–8 μm	-	-	[80]
ramon (*Brosimum alicastrum*)/seeds	Spherical	Average diameter 7–8 μm	C	15°, 17°, 18°, 20°, 23°, 26°,	[68]
Sago pith waste fibre	Temple bell-like oval	13–36 μm, mean diameter 28 μm	C	15.1°, 17.1°, 18.0°, 23.0°	[69]
Sago-kithul palm (*Caryota urens*)/flour	Oval	12–62 µm	A	15.1°, 17.1°, 17.9°, 23°	[70]
Sand rice (*Agriophyllum* *squarrosum*)/seeds	Spherical	Mostly < 1 μm	A	15.1°, 17.2°, 18.0°, 23.1°	[74]
Sand rice (*Agriophyllum* *squarrosum*)/seeds	Smooth spherical	1–2 µm, D [4,3] 1.1–1.3 µm	A	15.8°, 17.1°, 18.3°, 23.3°	[71]
Sweet potato/tubers	-	Bimodal; 0.8–3 µm and 6–40 µm	-	-	[90]
Sweet potato/tubers	-	Bimodal; 0.5–3 μm and 4–50 μm	C_A_	-	[18]
Talipot palm (*Corypha umbraculifera* L.)/flour	-	-	A	15.1°, 23.2°, doublet at 17.2°, 18.1°, 15.1°	[75]
*Treculia africana*/seeds	Irregular	Average diameter 20.6 µm	A	15.5°, 17.4°, 18.0°, 23.0°	[76]
Ulluco (*Ullucus tuberosus Caldas*)/tubers	Oval and conical	-	B	5.6°, 15°, 17°, 20°, 22°, 24°	[77]
Ulluco (*Ullucus tuberosus Caldas*)/tubers	Ellipsoid, oval, smooth, irregular	-	-	-	[91]
Ulluco (*Ullucus tuberosus Caldas*)/tubers	Elongated and rounded	-	B	5.6°, 15°, 17°, 20°, 22°, 24°	[92]
Winter squash (*Cucurbita maxima* Duch.)/fruit	Polyhedral, irregular	D [4,3] 33–45 µm	B	5.6°, 15°, 17.2°, 19.8°, 22.3°, 24.0°	[66]

**Table 3 foods-12-03794-t003:** Chain lengths in amylopectin.

Starch from	Polydispersity Index of Starch (Mw/Mn)	Chain Length Distributions in Amylopectin (%)				Reference
		DP 6–12	DP 13–24	DP 25–36	DP > 37	
Normal amaranth/seeds	-	28.6	49.9	12.6	9.0	[27]
Waxy amaranth	-	25.1	47.7	12.5	14.7	[95]
Waxy hull-less barley/kernel	-	27.2	51.4	14.8	6.6	[31]
Normal barley	-	20.8	48.9	17.7	12.6	[95]
Canna/green leaf	-	11.7	45.3	16.2	26.8	[95]
*Chlorella* sp. MBFJNU-17		23.4	26.0	13.6	31.9	[46]
Green banana	-	16.8	46.3	12.9	24.0	[95]
Lotus/root	-	16.4	47.2	15.4	21.0	[95]
High-amylose corn starch	-	8.5–9.7	40.7–43.9	20.3–21.3	26.1–29.5	[95]
Normal corn	-	20.1	50.2	17.6	12.1	[80]
Normal corn	-	17.9	47.9	14.9	19.3	[95]
Waxy corn	-	21.3	50.5	17.1	11.1	[27]
Waxy corn	-	17.0	49.4	17.1	16.5	[95]
Cat tail millet	-	20.2	53.8	12.7	13.3	[95]
Mung bean/seeds	-	22.7	48.7	17.5	11.7	[40]
Mung bean	-	15.6	47.6	18.3	18.5	[95]
Oat/seeds	-	26.8	47.4	16.9	8.9	[27]
Pumpkin, var. Miben	2.1	21.4	47.3	15.3	14.1	[66]
Quinoa/seeds	18.2	30.0	49.2	13.8	7.0	[71]
Quinoa/seeds	-	24.6	22.0	50.0	14.7	[27]
Quinoa/seeds		26.5	24.9	12.4	32.3	[46]
Normal rice/seeds	-	19.0	52.2	12.3	16.5	[95]
Sweet rice/seeds	-	23.5	48.7	13.7	14.0	[95]
Waxy rice/seeds	-	20.4	52.7	14.6	12.3	[27]
Waxy rice/seeds	-	27.4	53.4	12.6	6.6	[95]
Sand rice (*Agriophyllum* *squarrosum*)/seeds	1.3–5.8	30.8–39.8	45.9–53.1	9.4–11.8	2.0–4.9	[71]
Tapioca	-	17.3	40.4	15.6	26.7	[95]
Chinese taro	-	18.8	48.7	14.8	17.7	[95]
Water chestnut	-	17.8	43.7	15.3	23.2	[95]
Winter squash, var. Yinli	2.2	23.2	48.5	14.9	13.5	[66]
Winter squash, var. Heili	2.0	24.1	49.3	14.0	12.6	[66]

**Table 4 foods-12-03794-t004:** Thermal data of non-traditional starches.

Starch from	*To* (°C)	*Tp* (°C)	*Tc* (°C)	*Tc–To* (°C)	Δ*H_gel_* (J/g)	Δ*H_ret_* (J/g)	Reference
Acorn/fruits	41.7–45.2	78.4–84.7	117.4–118.9	73.7–76.3	17.7–23.4		[19]
Adzuki bean	51.4	73.3	90.1	38.7	8.1		[80]
Avocado/seeds	66.7–69.3	70.3–73.6	76.6–79.5	9.2–10.7	6.6–13.4		[98]
Achira starch	59.2	65.2	73.8	14.6	10.9		[9]
Amaranth/grains	59.4	68.4	82.3	22.9	17.4		[22]
Arrowroot starch	67.8	73.6	81.4	13.6	10.3		[4]
Waxy amaranth/seeds	62.4	69.3	75.9	13.5	10.2		[80]
Normal barley	52.0–61.4	58.1–65.3	62.7–74.4	-	6.8–14.2		[1]
Waxy barley	54.5–61.3	61.8–65.5	73.8–81.8	-	9.9–13.1		[1]
Amylose-barley	53.0–61.0	62.0–68.4	74.3–79.5	-	6.6–12.2		[1]
Bean (*Phaseolus vulgaris* L.)/seeds	54.6–69.4	64.7–76.8	74.1–81.3	-	3.8–16.3	5.8–9.0	[33]
Canna starch	59.0	63.6	72.1	13.1	9.4		[4]
Chestnut (*Castanea mollissima* BL.)/kernels	56.0	66.7	81.3	25.3	17.3		[43]
Chinese chestnut (*Castanea mollissima* BL.) var. Yanshan Zaofeng/fruit	57.4	65.6	75.1	17.7	9.6		[88]
*Chlorella* sp. MBFJNU-17	63.0	66.3	74.1	11.1	-		[46]
*Chrysophyllum albidum* (African Star Apple)/kernel	75.8	80.9	86.1	10.3	15.6		[47]
*Carioca* beans	69.7–70.7	75.0–76.6	79.9–81.0	10.1–13.8	7.8–9.1	4.5–4.7	[37]
*Euryale ferox*/seeds	84.7	110.3	118.1	33.4	1576.0		[50]
Ginger (*Zingiber officinale* Roscoe.) starch/waste product	70.7	-	78.1	7.4	15.2		[52]
Bambara groundnut/germinated seeds	70.9	72.8	85.6	14.7	3.6		[32]
Jackfruit/seeds	72.6–84.9	77.7–88.0	85.7–96.6	-	7.9–16.6		[55]
Lily (*Lilium* spp.)/bulb	61.3	64.5	72.7	11.4	15.7		[56]
Normal corn starch	64.0	69.7	74.4	10.4	12.8		[80]
Waxy corn starch	63.9	71.4	76.8	12.9	12.9		[80]
Mango/kernels	74.0	117.5	182.1	108.1	148.6		[60]
Mung bean/seeds	57.6	67.7	77.7	20.1	7.8		[40]
Mung bean/seeds	58.9	63.5	85.6	26.7	10.9		[41]
Mung bean/starch	58.6	64.6	73.0	14.4	10.5		[85]
Oat/seeds	55.5	60.6	65.7	10.2	8.0		[80]
Oat/seeds	51.1–69.5	57.8–66.8	-	-	8.6–14.6		[1]
Quinoa/seeds	56.3	64.5	74.0	17.7	16.1	6.2	[71]
Quinoa/seeds	52.7	61.2	69.5	16.8	9.4		[80]
Quinoa/seeds	61.1	66.3	73.9	12.8	14.6		[99]
Quinoa/seeds	54.8	60.1	67.3	12.5	8.1		[46]
*Phoenix sylvestris*/root tuber	68.2	82.3	95.0	26.8	11.3	1.6–2.9	[64]
Ramon (*Brosimum alicastrum*)/seeds	70.8	76.2	81.4	10.6	12.6		[68]
Sago pith waste fiber	142.9	149.6	167.1	24.2	270.2		[69]
Sand rice (*Agriophyllum* *squarrosum*)/seeds	70.9	75.1	79.3	8.4	10.9		[74]
Sand rice (*Agriophyllum* *squarrosum*)/seeds	67.5–68.1	72.8–74.2	81.4–84.1	13.8–16.3	19.8–24.7	11.6–11.9	[71]
Sweet potato/tubers	51.1–62.9	61.2–77.2	77.3–84.0	17.9–28.1	11.7–14.3		[90]
Pumpkin (*Cucurbita moschata* Duch. ex Poir.) cultivar Miben/fruit	66.9	71.6	78.0	12.8–32.2	16.89		[66]
Waxy rice/seeds	74.8	80.0	88.8	11.1	23.9		[80]
Winter squash. var. Heili/fruit	61.3	64.8	71.0	8.5	10.4		[66]
Winter squash. var. Yinli/fruit	65.3	67.9	73.8	14	13.9		[66]

*To*—onset temperature, *Tp*—peak temperature, *Tc*—conclusion temperature, *Tc–To*—gelatinization range, Δ*H_gel_*—enthalpy change during gelatinization, and Δ*H_ret_*—enthalpy change during gelatinization.

**Table 5 foods-12-03794-t005:** Pasting properties of starches.

Starch from	Starch Concentration	PV (mPa.s)	BD (mPa.s)	FV (mPa.s)	SB (mPa.s)	Peak Time (min)	Pasting Temperature (°C)	Reference
Acorn/kernels	12% (*w*/*v*)	3410	1160	3557	1307	-	74.6	[20]
Waxy amaranth/seeds	7% (db *w*/*w*)	842	295	639	92	-	-	[80]
Avocado/fruit	16.67% (*w*/*w*)	21820	-	6791	-	-	-	[83]
Black bean/seeds	10% (*w*/*v*)	1981	-	1753	-	-	-	[35]
Turkish beans	10.71% (*w*/*w*)	4430	-	5259	2592	-	71.8	[99]
Bean (*Phaseolus vulgaris* L.)/seeds	3% (db *w*/*w*)	667–4730	1–2165	1144–5170	437–2667	-	-	[33]
*Carioca* beans	8% (*w*/*w*)	1231–2218	80–346	1790–3556	640–1465	-	78.4–82.6	[37]
*Tartary* buckwheat/flour	10.71% (*w*/*w*)	3395	952	4129	1671		75.9	[34]
Chestnut	-	528	19	651	142	-	-	[87]
Chinese chestnut (*Castanea mollissima* BL.) var. Yanshan Zaofeng/fruit	12% db (*w*/*v*)	1796	445	2260	909		74.2	[88]
*Chrysophyllum albidum* (African Star Apple)/kernel	12% (*w*/*w*)	2959	526	3637	1204	5.9	81.6	[47]
*Euryale ferox*/seeds	-	5448	1656	5274	1482	3.9	82.3	[50]
Foxnut (*Euryale ferox* Salisb.)/kernels	10.71% (*w*/*w*)	1953–2734	52–249	2422–3078	521–593		81.8–88.7	[16]
Ginger (*Zingiber officinale* Roscoe.) starch/waste product	10% (*w*/*w*)	1550	10	3110	1560	-	89.6	[52]
Bambara groundnut/germinated seeds		4538–4813	446–485	5512–5891	1420–1563	7.1–7.4	87.1–88.2	[32]
Jackfruit/seeds	-	840–3800	110–1300	1350–5200	550–2400	-	64.2–91.3	[55]
Normal corn starch	7% (db *w*/*w*)	1166	162	1137	234	-	-	[80]
Normal corn starch	10.71% (*w*/*w*)	2872	-	2697	1038	-	75.5	[99]
Normal corn starch	10.71% (*w*/*w*)	2289	-	848	2763	-	-	[61]
Waxy corn/starch	7% (db *w*/*w*)	1912	1040	1011	139	-	-	[80]
Mango/kernels	8% (*w*/*w*)	1198	85	2139	1026	-	84.5	[60]
Kutki millet/grain	10.71% (*w*/*w*)	2580	1426	3047	1893			[61]
Mung bean/seeds	10.71% (*w*/*w*)	5167	2260	4355	1425	-	75.1	[40]
Oat/seeds	7% (db *w*/*w*)	1207	488	2039	1319	-	-	[80]
*Phoenix sylvestris*/root tuber	10% db (*w*/*v*)	1824	-	2887	-		71.1	[64]
Pumpkin (*Cucurbita moschata* Duch. ex Poir.) cultivar Miben/fruit	10% db (*w*/*v*)	4468	1352	4075	959	4.4	75.2	[66]
Quinoa/seeds	7% (db *w*/*w*)	909	137	965	193	-	-	[80]
Quinoa/seeds	10.71% (*w*/*w*)	3061	2125	3038	2102	4.0	68.7	[71]
Quinoa/seeds	10% (db *w*/*v*)	4800	1500	5500	2100		69.0	[99]
Sand rice (*Agriophyllum* *squarrosum*)/seeds	6% (db *w*/*w*)	2458	885	3864	2853		75.5	[74]
Sand rice (*Agriophyllum* *squarrosum*)/seeds	10.71% (*w*/*w*)	1474–1601	200–522	3033–3426	1868–2338	4.7–5.5	78.3–79.2	[71]
Sweet potato/tubers	10% (*w*/*v*)	4047	-	2977	853	-	76.7	[99]
Sweet potato/tubers	(10% *w*/*v*)	3918–4929	1479–2591	2769–3579	541–769	4.1–4.6	70.1–78.7	[90]
Ulluco (*Ullucus tuberosus Caldas*)/tubers	7% (db *w*/*w*)	8600	-	3470	570	-	62.8	[77]
Waxy rice/seeds	10% (db *w*/*w*)	1407	599	1018	210	-	-	[80]
Winter squash. var. Heili/fruit	10% (db *w*/*w*)	4690	430	6073	1814	5.8	70.6	[66]
Winter squash. Var. Yinli/fruit	10% (*w*/*v*)	6266	2390	4664	787	4.3	72.3	[66]

PV—peak viscosity, FV—final viscosity, BD—breakdown, and SB—setback.

**Table 6 foods-12-03794-t006:** Swelling power, solubility, and water absorption.

Starch from	P (g/g)		SW (%)		WAI (g/g)		Reference
	60 °C	80 °C	60 °C	80 °C	60 °C	80 °C	
Acorn/fruits	3.0	10.1	0.2	5.4	-	-	[19]
Acorn/kernels	3	12	2	6	-	-	[20]
Anchote tuber	5.0	4.6	13.2	13.8	3.9	3.9	[24]
Annatto seeds	6.2	18.0	3.8	15.0	-	-	[25]
Avocado/seeds	3.5–6.5	7.0–10.5	0.5–17.5	5.5–17.0	3.5–4.8	6.0–9.8	[98]
Arrowroot starch	-	7.5	-	7.0	-	2.3	[82]
Black bean/seeds	4	12	4	48	-	-	[35]
Canistel/seeds	2.7	10	2.6	9.0	-	-	[36]
Canna starch	14.4	32.2	13.3	20.0	13.4	31.2	[4]
*Euryale ferox*/seeds	-	-	1	9	-	-	[50]
Foxnut (*Euryale ferox* Salisb.)/kernels	0.8	4.3–5.8	0.3	2–2.7	-	-	[16]
Ginger (*Zingiber officinale* Roscoe.) starch/waste product	-	4.5	-	3.7	-	-	[52]
Jackfruit/seeds	3–6	7.5–13	4.2–6.2	8.2–13.8	-	-	[55]
Lily (*Lilium* spp.)/bulb	-	8.6	-	6.6	-	-	[56]
Normal corn starch	4.1	9.7	-	-	-	-	[61]
Mango/kernels	8.6	-	6.6	-	-	-	[60]
Kutki millet/grain	3.8	8.3	-	-	-	-	[61]
Mung bean/seeds	2.9	9.3	1.0	8.1	-	-	[40]
*Phoenix sylvestris*/root tuber	8	15	22	32	-	-	[64]
Unripe plantains (*Musa* sp. AAB)	4.4	-	1.6	-	4.2	-	[65]
Quinoa/seeds	10	21	4	22	-	-	[71]
Sand rice (*Agriophyllum* *squarrosum*)/seeds	2.5	13–15	1	7–8	-	-	[71]
*Treculia africana*/seeds	3.9	12	-	-	-	-	[76]

**Table 7 foods-12-03794-t007:** Rapidly digestible, slowly digestible, and resistant starch.

Starch from	RDS (%)	SDS (%)	RS (%)	Reference
Acorn/kernels	16.2, 69.6 *	37.6, 14.5 *	46.2, 15.9 *	[20]
Adzuki beans (*Vigna angularis* L.)	27.6	30.6	40.8	[80]
Amaranth/grains	57.3	32.8	9.9	[22]
Annatto seeds	23	37	14	[25]
Waxy hull-less barley	55.0	2.0	43.0	[31]
Kidney bean *	82.2	4.5	13.3	[84]
*Carioca* beans	4.2–5.2	6.5–11.8	83.0–88.7	[37]
*Tartary* buckwheat/flour	57.3	30.1	12.6	[34]
Chestnut	21.1	21.3	57.6	[87]
Chinese chestnut (*Castanea mollissima* BL.) var. Yanshan Zaofeng/fruit	8.2	33.6	58.2	[88]
*Chlorella* sp. MBFJNU-17	48.8, 88.0 *	8.8, 3.5 *	42.4, 8.5 *	[46]
*Euryale ferox*/seeds	26.0	37.5	36.5	[50]
Foxnut (*Euryale ferox* Salisb.)/kernels	8.4–14.0, 79.1–83.5 *	22.8–26.4,10.6–11.6 *	59.6–68.8, 4.9–10.3 *	[16]
Ginger (*Zingiber officinale* Roscoe.) starch/waste product	1.8, 39.85 *	11.7, 23.01 *	63.1, 8.50 *	[52]
Gorgon nut/seed	-	47.0–60.1 *	-	[53]
Bambara groundnut/germinated seeds	12.1–12.9	14.7–15.8	72.0–73.2	[32]
Lily (*Lilium* spp.)/bulb	1.8, 20.3 *	8.2, 27.5 *	90.0, 52.2 *	[56]
Lotus/seed	18.5	16.0	65.5	[89]
Unripe plantains (*Musa* sp. AAB)	1.2, 58.8 *	8.7, 15.9 *	90.0, 25.2 *	[65]
Pumpkin, var. Miben	2.5, 78.1 *	4.9,3.1 *	92.7,18.8 *	[66]
Quinoa/seeds	72 *	20 *	8 *	[71]
Quinoa/seeds	65.6, 85.0 *	1.9, 6.5 *	32.5, 8.5 *	[46]
Himalayan rice/seeds	-	-	0.4–2.3	[72]
Himalayan rice/seeds			85.4–92.8	[73]
Sand rice (*Agriophyllum* *squarrosum*)/seeds *	66–70	24	8–12	[71]
Sand rice (*Agriophyllum* *squarrosum*)/seeds	89.3	4.1	8.6	[74]
Sweet potato/tubers	52.9	6.6	38	[111]
Sweet potato/tubers	2.2–1.9, 70.5–81.4 *	8.8–30.9, 2.1–11.0 *	58.2–89.1, 10.8–23.3 *	[90]
Winter squash. var. Heili/fruit	4.5, 90.5 *	9.6, 1.3 *	85.9, 8.2 *	[66]
Winter squash. var. Yinli/fruit	2.6, 84.2 *	4.9, 4.1 *	92.8, 11.6 *	[66]

* cooked.

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
