# Peer review of "Non-Traditional Starches, Their Properties, and Applications"

_foods, 2023, doi:10.3390/foods12203794_

Round 1

Reviewer 1 Report

1.      The abstract should be updated to show the knowledge gathered from the review and the significance of the results at the end.

2.      Line 16. Delete “, presented in the form of tables,”

3.      Table 1. The chemical concentrations utilized are required.

4.      Line 63-73. What conclusions can you draw from Table 1? Please brief in this section.

5.      In Section 3. Size, Shape, and Crystallinity of Starch Granules. Images of starch granules from various raw materials are required. 

6.      Section 9 and 10. Please provide some examples of FTIR/Raman/NMR spectra.

7.      Section 11. Potential Application of Non-traditional Starches. Please offer information on the feasibility or usability of these starches for plant-based products, which are now trendy.

8.      Because the authors have addressed modified starch throughout the article, a section titled "Starch Modification" is required. Mention the method of modification as well as the characteristics of the resulting modified starch before recommending the best/possible methods for your non-traditional starches.

9.      The review should include some traditional/indigenous rice starch, particularly from Asian countries.

10.   A concise and informative conclusion is required. As a result, please rewrite the conclusion. Please also describe the future trend of utilization of these starches as well as their industrial limitations.

Extensive editing of English language required.

Author Response

see in eclosure

Reviewer 2 Report

This paper focuses on isolation/modification and different properties of novel starch sources from accessible botanical origins. As a novelty of this research, is the use of waste for isolation to obtain a new type of starch. Different properties of starches including chemical structure, size and crystallinity of granules, thermal and pasting properties, swelling power, and digestibility have been reported in this paper. The paper benefits from novel characterizing techniques to evaluate starch properties including laser diffraction, X-ray diffraction, differential scanning calorimetry, rheological measurement, enzyme procedures to analyze digestibility, FTIR, Raman and NMR spectroscopies.

1)     The abstract should be revised in order to express the results in a more appropriate way. Please include the core results of your study and the most important results with content in the abstract. Statistical evaluation in your results should be seen.

2)     Please define your research hypothesis at the end of your introduction. Also point out the novelty of this research compared to existing literature.

3)     There are too many paragraphs in Introduction. You may combine them in a coherent structure.

4)     Since authors have outlined different methods for the extraction of Starch from plant sources, I would advise to not forget a novel dry-based separation methods of starch reported earlier. (i.e., you may use:

Migration of gluten under shear flow as a novel mechanism for separating wheat flour into gluten and starch: https://doi.org/10.1016/j.jcs.2007.10.005)

5)      Conclusion section also comprises many short paragraphs that can be combined in a cohesive structure.

Author Response

see in enclosure

Round 2

Reviewer 1 Report

The replies were made to be the acceptable rationale point-by-point and the adjustment was made in accordance. It is therefore acceptable.

Minor editing of English language required